# Variational Amodal Object Completion

**Huan Ling**[1,2,3]    **David Acuna**[1,2,3]    **Karsten Kreis** [1]

**Seung Wook Kim**[1,2,3]    **Sanja Fidler**[1,2,3]

NVIDIA[1]    University of Toronto[2]    Vector Institute[3]

{huling ,dacunamarrer, kkreis, seungwookk, sfidler}@nvidia.com

## Abstract

In images of complex scenes, objects are often occluding each other which makes perception tasks such as object detection and tracking, or robotic control tasks such as planning, challenging. To facilitate downstream tasks, it is thus important to reason about the full extent of objects, i.e., seeing behind occlusion, typically referred to as *amodal instance completion*. In this paper, we propose a variational generative framework for amodal completion, referred to as Amodal-VAE, which does not require any amodal labels at training time, as it is able to utilize widely available object instance masks. We showcase our approach on the downstream task of scene editing where the user is presented with interactive tools to complete and erase objects in photographs. Experiments on complex street scenes demonstrate state-of-the-art performance in amodal mask completion, and showcase high quality scene editing results. Interestingly, a user study shows that humans prefer object completions inferred by our model to the human-labeled ones.

## 1   Introduction

One of the most remarkable properties of the human visual system is the ability to rapidly recognize objects and understand their spatial extent in complex visual scenes, even when objects are barely visible due to occlusion [9, 42]. This is important, as it allows humans to more accurately anticipate what can happen a few moments into the future, and plan accordingly. We expect such a capability to also benefit robotic systems. Reasoning about objects and their extent is also key in other contexts, for example, in semantic image editing tasks. Imagine a user that wants to erase an object from a photograph, and possibly even manipulate objects that are partially hidden behind it. To do this, an A.I. system needs to be able to "complete" the occluded objects in the scene, both in their spatial extent, i.e., their masks, as well as in appearance. This problem is typically referred to as *amodal instance completion*, and is an important component of many applications.

However, most research in the domain of semantic segmentation, has focused on the "modal" perception of the scene [6, 11, 34], i.e., segmenting visible pixels of the objects, for which large-scale annotated datasets are available [7, 23, 41]. The lack of labeled data for amodal segmentation is likely due to the difficulty and ambiguity of the annotation task. Amodal annotation of occluded objects requires a human labeler to draw an imagined contour rather than tracing a visible contour in an image, which requires drawing skills that not all annotators possess. In cases where objects are highly occluded there may also be multiple valid hypotheses for a plausible completion.

In this work, we propose a variational generative framework for amodal instance completion, called Amodal-VAE. It does not require amodal labels at training time, and exploits instance masks of visible parts of the objects that are widely available in current datasets. Our approach learns to reconstruct full objects from partial masks by training a variational autoencoder in carefully designed

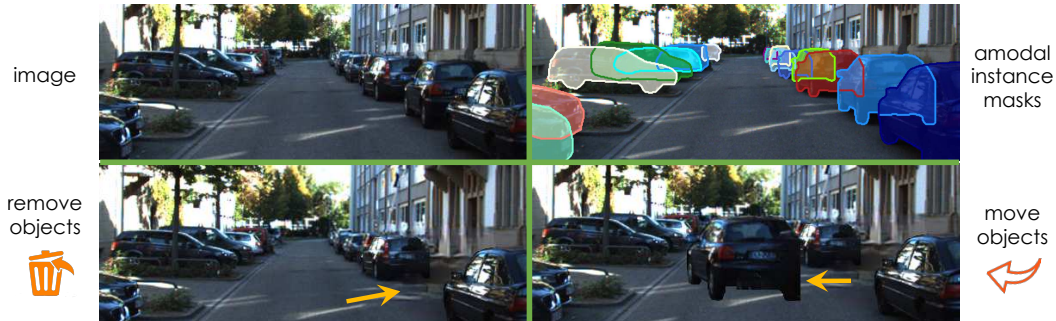

Figure 1: We present a new method for amodal object completion (top), and showcase our work on scene editing (bottom). User is presented with interactive tools to complete, erase, and manipulate objects in an image.

stages that allow us to model the complete mask with a low-dimensional latent representation. The probabilistic framework naturally incorporates the ambiguity in the mask completion task, and is able to produce multiple plausible completions which existing work cannot. We showcase our approach on the downstream task of scene editing where the user is presented with interactive tools to complete and erase objects in an image. Experiments demonstrate significant improvements over the recently released state-of-the-art approach [39]. A user study further reveals that participants strongly prefer amodal masks produced by our model over the human-annotated amodal masks.

## 2    Related Work

We focus our review on amodal mask completion which is the primary contribution of our work. The task of amodal instance segmentation aims at segmenting both visible and occluded parts of an object instance. This is in contrast to traditional semantic segmentation [34, 6] or instance segmentation tasks [11, 28, 1, 24], which aim to segment only the visible pixels of an object. Prior work usually decomposes amodal instance segmentation into instance segmentation and amodal mask completion. Supervision is needed for both stages, typically resorting to either synthetic datasets or human-provided labels, which we discuss below.

**Real Datasets:**    Recently, human-labeled real datasets have been collected for amodal instance segmentation. Authors extended KITTI [10] to create KINS [27], and COCO [23] to create COCOA [42]. However, there is little available labeled data, in part due to the ambiguity of the labeling task.

**Synthetic Datasets:**    One plausible way to get amodal labels is to exploit graphics renderers [14, 43, 17]. In [14], a photo-realistic video dataset is extracted from the GTA-V game along with pixel-accurate masks. In [16], 3D models are aligned with images from PASCAL 3D+ [36] and rendered along with their annotated 3D pose to obtain masks and amodal bounding boxes. In [8], the authors created DYCE by taking snapshots from 3D synthetic scenes [43]. While 3D content provides labels for "free" via rendering, it is not widely available and typically lacks diversity and realism.

**Simulated Data:**    A simple way to utilize real data annotated with instance (but not amodal) masks is by simulating occlusion, i.e., by overlaying objects on top of other objects [21, 37, 39]. One problem with this type of approach is that the composited images do not look natural and thus appearance-based models may not generalize well to real images. [37] created OVD (Occluded Vehicle Dataset) by randomly placing pedestrians and vehicles on base images and exploited the Deep Harmonization [35] technique to make synthetic images look natural. Our work, while also relying on occlusion simulation, does so only for object masks, ignoring appearance altogether. Our method can thus exploit either rendered masks, or masks from one dataset for use on another dataset.

**Methodology:**    In most prior work, labels, either real or synthetic, are used in a standard supervised framework. In [21], the authors perform amodal segmentation by iteratively expanding the bounding box around an instance mask based on heat intensity. [27] proposes an occlusion classification branch on top of RPN [28]. In addition to the standard mask prediction loss, [37] utilizes a discriminator loss to encourage amodal predictions to look more similar to amodal masks rendered via the Shapenet dataset [5]. Our work also bears similarity to the recent De-occlusion paper [39] due to the application to scene editing. However, the approaches for mask completion differ in methodology, where ours frames the problem probabilistically, while [39] is a deterministic method.

Related to our work is [32, 38], where the authors train a VAE [20] to learn a 3D shape prior. This prior is then used to generate closed 3D meshes [32] from partial point cloud observations.

Several unsupervised methods have also been proposed for amodal mask completion. Prior works treat amodal completion as a contour completion problem, usually recovered by minimizing shape energy. [18] uses Euler spirals, [30] exploits Contour-Completion Random Fields and [22] utilizes minimum Hamiltonian cycles and Bezier curves. However, most of these unsupervised methods focus on simple shapes and cannot easily be scaled to real world datasets.

## 3 Background and Problem Formulation

In this section, we review Variational Autoencoders (VAEs) and we formally define the problem of amodal instance completion, which we are addressing.

### 3.1 Variational Autoencoders

Given a dataset $\mathcal{D} = \{\boldsymbol{y}_i\}_{i=1}^N$, the VAE framework enables us to learn a latent variable generative model $p(\boldsymbol{y}, \boldsymbol{z}) = p_{\boldsymbol{w}_1}(\boldsymbol{y}|\boldsymbol{z})p(\boldsymbol{z})$, where $p(\boldsymbol{z})$ is a prior distribution over latent variables and $p_{\boldsymbol{w}_1}(\boldsymbol{y}|\boldsymbol{z})$ is a likelihood distribution, usually interpreted as a decoder and typically parametrized by a neural network with parameters $\boldsymbol{w}_1$ [20, 29]. Since the true posterior distribution $p(\boldsymbol{z}|\boldsymbol{y})$ is intractable, VAEs employ an auxiliary approximate posterior distribution or encoder $q_{\boldsymbol{w}_2}(\boldsymbol{z}|\boldsymbol{y})$, parametrized by another neural network with parameters $\boldsymbol{w}_2$. When additional information about the data is available, such as the samples' classes or categories $\boldsymbol{c}$, the framework can be extended to conditional VAEs, in which the encoder, prior and decoder can be conditioned on this class information [19, 31].

VAEs are trained via variational inference, maximizing the Evidence Lower BOund (ELBO). Here, we consider the case in which only the encoder is conditioned on additional class information $\boldsymbol{c}$ that is available for all samples in the dataset $\mathcal{D}$. The ELBO then is

$$\mathcal{L}_{\text{VAE}}(\boldsymbol{w}_1, \boldsymbol{w}_2) = \mathbb{E}_{\boldsymbol{y}, \boldsymbol{c} \sim \mathcal{D}} \left[ \mathbb{E}_{\boldsymbol{z} \sim q_{\boldsymbol{w}_2}(\boldsymbol{z}|\boldsymbol{y}, \boldsymbol{c})}[\log p_{\boldsymbol{w}_1}(\boldsymbol{y}|\boldsymbol{z})] - \lambda \, D_{\text{KL}}(q_{\boldsymbol{w}_2}(\boldsymbol{z}|\boldsymbol{y}, \boldsymbol{c}) \| p(\boldsymbol{z})) \right] \quad (1)$$

When calculating gradients during training, the expectation over the data is estimated using mini-batches and the expectation over the latent variables $\boldsymbol{z}$ is usually calculated using a single sample from the approximate posterior. Parameter updates are done with stochastic gradient descent, employing the re-parameterization trick [20, 29]. Due to the KL-regularization, the model learns to encode data $\boldsymbol{y}$ in an efficient low-dimensional latent representation $\boldsymbol{z}$. Although strict variational inference corresponds to $\lambda = 1$, it has been shown that different values of $\lambda$ allow us to carefully control the balance between the KL and the reconstruction terms [3, 40, 2, 13, 4], which can be beneficial.

### 3.2 Amodal Instance Completion

Let $\mathcal{D} = \{\hat{\boldsymbol{y}}_i\}_{i=1}^N$, be a dataset of "partial" instance object masks $\hat{\boldsymbol{y}}_i \in \hat{\mathbb{Y}}$ in images. We can define an *Amodal Mask Completion* method as a mapping $\boldsymbol{f} : \hat{\mathbb{Y}} \to \mathbb{Y}$ with completed masks $\boldsymbol{y}_i \in \mathbb{Y}$. In words, the amodal instance completion task recovers the occluded part of a particular object from the partially occluded instance mask. If available, we can use additional information in the function $\boldsymbol{f}$, such as the images' RGB pixel values or the instances' classes $\boldsymbol{c}_i$, like in the VAE framework. Note that, formally, the set of realistic complete masks $\mathbb{Y}$ is a subset of all possible partial masks $\hat{\mathbb{Y}}$.

## 4 Variational Object Completion

A trivial solution to the task of Amodal Mask Completion would be collecting a training dataset $\mathcal{D}_{\text{train}} = \{\boldsymbol{y}_i, \hat{\boldsymbol{y}}_i\}_{i=1}^N$ consisting of *paired* partial masks $\hat{\boldsymbol{y}}_i$ and corresponding complete masks $\boldsymbol{y}_i$ (and potentially additional information, such as instance classes $\boldsymbol{c}_i$). Then, we could fit a parametric model, i.e. a neural network, to it by treating it as an image segmentation problem. However, annotating an amodal dataset is challenging, time-consuming, expensive and sometimes ambiguous, as objects resulting from occlusions may not even be well-defined. The resulting annotations may vary from individual to individual, which could also make learning more difficult. Instead, we exploit a weakly-supervised approach, where we have access to data with only partially visible masks ($\hat{\mathbb{Y}}$) and *separate* data with only full masks ($\mathbb{Y}$). As shown in Figure 2, we are using a VAE framework, in which we first encode partially visible masks $\hat{\boldsymbol{y}}$ into a smooth latent space and then decode the resulting latent codes $\boldsymbol{z}$ into the full masks $\boldsymbol{y}$.

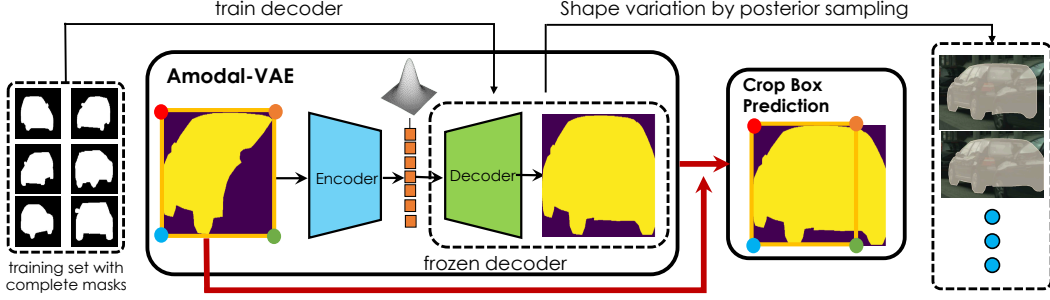

**Figure 2: Amodal-VAE**: We first encode partially visible masks $\hat{y}$ into a low-dimensional latent space and then decode the latent code $z$ into the full mask $y$ and resize it. Furthermore, we can sample different latent codes from the approximate posterior distribution. These samples correspond to different plausible mask completions.

A crucial advantage of the probabilistic VAE-based framework is that it naturally captures the ambiguity when completing partial masks in its posterior distribution (see Fig 6). Furthermore, it also deals gracefully with inputs that it is uncertain about. Since the model is trained such that all points under the prior distribution map to realistic completed masks, slightly erroneous latent code predictions still decode into well-defined outputs. We denote our model as Amodal-VAE. Next, we present our Amodal-VAE and how we train it in order to overcome the previously discussed challenges in more detail.

## 4.1   Learning to Reconstruct Full Objects

We start by presenting the high-level architecture of Amodal-VAE. For simplicity, we assume a factorial Normal prior distribution $p(z) \sim \mathcal{N}(0, \mathcal{I})$ and factorial Normal approximate posteriors $q_{w_2}(z|y, c)$ and $\hat{q}_{w_3}(z|y, c)$ with means and standard deviations parametrized via convolutional neural networks that also see the objects' categories $c$, which are available in all datasets we are working with or can be predicted if necessary. The decoder $p_{w_1}(y|z)$ is a factorial Bernoulli distribution, predicts binary masks, and is parametrized using a deconvolutional neural network (see supplementary material for details). To best leverage the two separate datasets $\mathbb{Y}$ with fully visible masks and $\hat{\mathbb{Y}}$ with partially visible masks, we train Amodal-VAE in three stages.

**(1) Full-Mask-only Training:**   We want Amodal-VAE to generate only realistic full masks, even when provided with partial masks that are significantly occluded as input. Hence, during the first step we focus on learning the generative component $p_{w_1}(y|z)p(z)$ of the model and we train Amodal-VAE on full masks only. Amodal-VAE is trained using the ELBO defined in Eq. 1 on $\mathbb{Y}$. It learns low-dimensional representations of complete masks of real objects in its continuous latent space.

**(2) Simulated Partial-to-Full-Mask Training:**   After (1), any point in latent space under the prior maps to a realistic completed mask. Now, based on the full mask data, we simulate various occlusions, hence generating a synthetic dataset of paired partial and complete masks of the form $\mathcal{D}_{\text{train}} = \{y_i, \hat{y}_i\}_{i=1}^{N}$. Freezing the previously learnt decoder, i.e. the decoder $p_{w_1}(y|z)$, we then learn a new encoder $\hat{q}_{w_3}(z|\hat{y}, c)$ with parameters $w_3$ that maps partial masks $\hat{y}$ to points in latent space $z$ that decode into the correct completed masks $y$.

For constructing the synthetic dataset, we sample random instances $y_{\text{foreground}}$ and $y_{\text{instance}}$ from $\mathbb{Y}$ and mask out $y_{\text{instance}}$ by randomly positioning $y_{\text{foreground}}$ in front of it, similar to [39]. We can now maximize the following adapted ELBO objective

$$\mathcal{L}_{\text{Amodal-VAE}}(w_3) = \mathbb{E}_{\hat{y}, y, c \sim \mathcal{D}_{\text{train}}} \Big[ \mathbb{E}_{z \sim \hat{q}_{w_3}(z|\hat{y}, c)}[\log p_{w_1}(y|z)] - \lambda D_{\text{KL}}(\hat{q}_{w_3}(z|\hat{y}, c) \| p(z)) \Big] \quad (2)$$

where $\hat{y}$ are the simulated partial masks, $y$ are the full masks, and $c$ is additional object class information. Notice that only the new encoder parameters $w_3$ are optimized and that we do not use the RGB image information.

The composition of the new encoder with the frozen decoder forms the *amodal instance completion mapping*, which we can formally express as $f(\hat{y}, c) = p_{w_1}^t(\hat{q}_{w_3}^\mu(\hat{y}, c))$, where we defined the deterministic functions $\hat{q}_{w_3}^\mu(\hat{y}, c)$ as the mean of $\hat{q}_{w_3}(z|\hat{y}, c)$ and $p_{w_1}^t(z)$ as the binary output mask calculated from pixelwise Bernoulli probabilities $p_{w_1}(y|z)$ with threshold $t$.

Intuitively, the first term in Equation 2 is the reconstruction loss that guides the encoder to find an

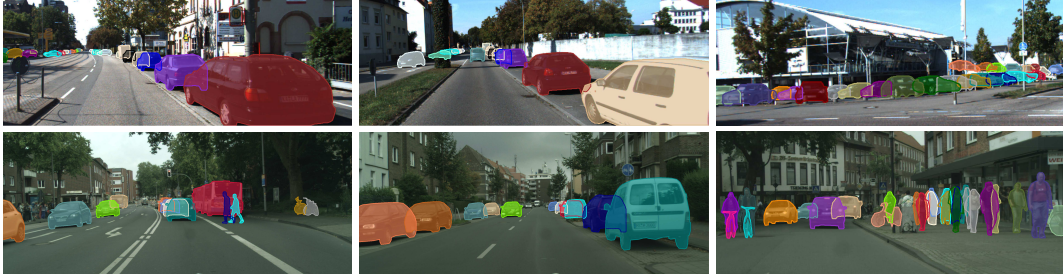

Figure 3: Qualitative results of amodal completion. Top: Results on KINS. Bottom: Results on Cityscapes.

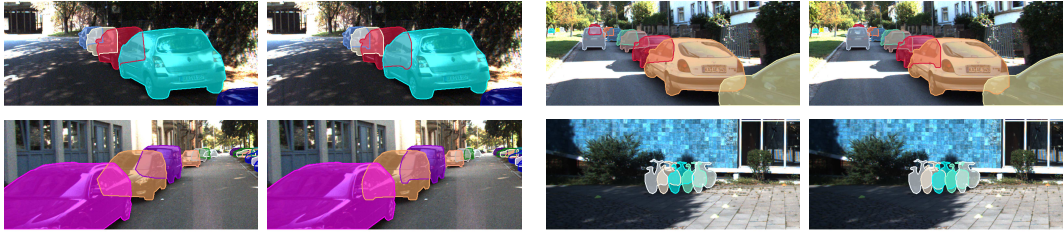

| GT | Our Prediction | GT | Our Prediction |

Figure 4: Results of Amodal-VAE vs human-annotated amodal masks. Results are shown on KINS test set.

appropriate position in the low dimensional Gaussian manifold which is decoded to $\mathbb{Y}$. The second term, the KL loss, regularizes the new approximate posterior $\hat{q}_{\boldsymbol{w}_3}(\boldsymbol{z}|\hat{\boldsymbol{y}}, \boldsymbol{c})$ to generate only encodings that fall under the prior distribution $p(\boldsymbol{z})$. Because of the first training step and since we keep the decoder frozen, all such encodings $\boldsymbol{z}$ map to complete masks.

To aid the new encoder to more easily search the latent space, we exploit an additional latent code distance loss. We pull encodings from complete and corresponding partial masks close to each other, since they both need to decode into the same full masks. We minimize the following loss:

$$\mathcal{L}_{\text{LatentCode}}(\boldsymbol{w}_3) = \mathbb{E}_{\hat{\boldsymbol{y}}, \boldsymbol{y}, \boldsymbol{c} \sim \mathcal{D}_{\text{train}}} \left[ \mathbb{E}_{\hat{\boldsymbol{z}} \sim \hat{q}_{\boldsymbol{w}_3}(\boldsymbol{z}|\hat{\boldsymbol{y}}, \boldsymbol{c}), \boldsymbol{z} \sim \hat{q}_{\boldsymbol{w}_3}(\boldsymbol{z}|\boldsymbol{y}, \boldsymbol{c})} \frac{1}{2} \left[ \hat{\boldsymbol{z}} - \boldsymbol{z} \right]^2 \right], \quad (3)$$

for paired $\hat{\boldsymbol{y}}$ and $\boldsymbol{y}$. We approximate the inner expectation using single samples from the approximate posteriors. We found adding this loss to the ELBO objective to slightly increase performance. However we found that it can't replace the reconstruction loss. The final loss becomes:

$$\mathcal{L}(\boldsymbol{w}_3) = \mathcal{L}_{\text{LatentCode}}(\boldsymbol{w}_3) + \mathcal{L}_{\text{Amodal-VAE}}(\boldsymbol{w}_3) \quad (4)$$

**(3) Partial-Mask-only Finetuning:** In the third training stage, we "finetune" the Amodal-VAE by training its encoder in standard VAE-fashion using *only* partial masks from $\hat{\mathbb{Y}}$, masking out all non-visible pixels. Finetuning the Amodal-VAE in this way helps the model to deal with complex realistic occlusions, which may not occur during the occlusion simulation in (2), for example since we only use single foreground instances to create simulated occlusions. The decoder remains frozen. For a partially visible mask $\hat{\boldsymbol{y}}$, we define its visible pixels as $\hat{\boldsymbol{y}}^{\text{vis}}$. We can define an ELBO as

$$\mathcal{L}_{\text{Finetuning}}(\boldsymbol{w}_3) = \mathbb{E}_{\hat{\boldsymbol{y}}, \boldsymbol{c} \sim \hat{\mathbb{Y}}} \left[ \mathbb{E}_{\boldsymbol{z} \sim \hat{q}_{\boldsymbol{w}_3}(\boldsymbol{z}|\hat{\boldsymbol{y}}, \boldsymbol{c})} [\log p_{\boldsymbol{w}_1}(\hat{\boldsymbol{y}}^{\text{vis}}|\boldsymbol{z})] - \lambda \, D_{\text{KL}}(\hat{q}_{\boldsymbol{w}_3}(\boldsymbol{z}|\hat{\boldsymbol{y}}, \boldsymbol{c}) \| p(\boldsymbol{z})) \right] \quad (5)$$

where we consider only the reconstruction loss on the visible pixels.

In training stages (2) and (3), we additionally apply a spatial transformer network on the output, that learns to resize the completed masks such that they can be pasted back into the scene (see Sec 4.2).

**Motivation:** One may ask, why separate training stages (1) and (2)? When learning the actual amodal completion model in step (2), the approximate posterior sees different partial masks, which can look entirely different due to different simulated occlusions, but that nevertheless map to similar completed masks. Alternatively, similar partial masks may correspond to very different completed masks. Training on such data constitutes a very difficult and ambiguous learning problem, unlike regular VAE training. If the generative component, i.e. the decoder, was also trained like this, it would result in a weaker model encoding less information in latent space. Therefore, we found it to be beneficial to separately train the generative component in robust standard-VAE fashion with full masks only first and then freeze it. After all, we know that we want to generate only ever full masks.

In other words, we are separating the difficulty of learning a high quality generative component from the difficulty of learning to map many different partial masks to similar completed masks and vice versa. Note that we also have to train the spatial transformer in step (2). It is easier to first learn the decoder on full masks only and then separately learn the spatial transformer on top of the "correct" decoder, instead of training both simultaneously.

## 4.2 Resizing Completed Masks with Spatial Transformers

Both input and output of Amodal-VAE are tightly cropped 2D instance masks, separately resized or squeezed to the model's fixed input and output dimensions. Therefore, the output masks are not in the same scale as the partial input masks. Because of that, we cannot simply resize and paste the completed masks back into the image. To overcome this hurdle, we learn an affine transformation that shifts and scales the output mask to correct for the discrepancy. The output mask can then be pasted back into the full image using the resizing and positioning of the partial input mask (see Fig 2).

With an instance's partial mask $\hat{y}$ and completed mask $y$, generated by Amodal-VAE's decoder in the VAE's fixed output dimensions, we learn a spatial transformation function $g_\theta(y, \hat{y}) \to y'$ such that the transformed $y'$ is the completed mask in the same scale and at the same position as the input mask $\hat{y}$. Specifically, we first predict the transformation parameters

$$(t_x, t_y, s_x, s_y) = g_\theta(y, \hat{y}) \qquad A_\theta = \begin{bmatrix} s_x & 0 & t_x \\ 0 & s_y & t_y \end{bmatrix} \qquad (6)$$

where $g_\theta$ is a neural network and $A_\theta$ is a 2D affine transformation matrix that is applied to each pixel in $y$ and used to do differentiable image sampling as defined in [15]. The transformation defined through $g_\theta$ and $A_\theta$ is end-to-end differentiable and can be trained by backpropagation together with the Amodal-VAE. The spatial transformer function, operating on the Amodal-VAE output, is trained during training stages (2) and (3) (in training stage (1) we train on complete masks only).

## 5 Experiments

We now extensively evaluate our Amodal-VAE and show its application to interactive scene editing. Please refer to supplementary material for training and model implementation details.

**Dataset:** We focus on street scenes in this paper. **KINS** [27] is a large scale dataset derived from KITTI [10], which contains both instance and amodal annotations. The dataset consists of 7,474 images for training and 7,517 images for testing. There are 18,241 and 17,646 complete instances in the training and test set respectively. Following [39], we use the first $\approx 10\%$ images from the test set as validation set (750 images in total). In this paper, we only exploit instance masks in training and amodal ground truth labels are only used for evaluation. The **Cityscapes** dataset [7] contains 5,000 images of driving scenes, including 2,975 images for training, 500 for validation, and 1,525 for testing. In the training set, 11,251 out of 52,469 instances are without occlusion. The instance masks in Cityscapes are finely annotated for the visible portions of the objects, however, no amodal annotations are available. In this paper, we treat Cityscapes as an additional dataset to test generalization of our approach.

### 5.1 Amodal Mask Completion

**Comparisons:** We first benchmark our approach for the task of amodal mask completion. To compare with baseline models, we use the amodal completion setting introduced in [39], where at test time RGB images and ground truth (GT) instance masks are provided as input to our model. Since our model does not exploit specific foreground occlusion masks as input, we use the De-occlusion-NOG (no order grounding) setting as a baseline.

The performance of our model on KINS is shown in Table 1. Because occluded regions are relatively small compared to full masks, the input instance masks have a high 87.03% mean Intersection over Union (mIOU) with the GT full masks. For this reason, we separately evaluate mIOU on the invisible area only as well. Results show that Amodal-VAE outperforms the state-of-the-art De-occlusion [39] model by 5.66% for invisible mIOU and 0.64% for full mIOU, which is a significant improvement.

For another baseline experiment, we generate a synthetic dataset from the KINS training set. Using the full mask data, for each mask we simulate 5 different occlusions by randomly pasting another mask as foreground, hence generating a synthetic dataset of paired partial and complete masks consisting of 91,205 examples. We can now use a nearest neighbor-based approach for mask completion. We

| Method | GT Crop | mIOU | Invis. mIOU |
|---|---|---|---|
| Instance Mask | ✗ | 87.03 | 0 |
| Nearest Neighbor Mask | ✗ | 93.71 | 54.97 |
| De-occlusion | ✗ | 94.04 | 57.19 |
| Amodal-VAE | ✗ | **94.68** | **62.85** |
| RGB-Amodal-VAE | ✗ | 94.53 | 61.97 |
| Amodal-VAE + GT Box | ✓ | 97.64 | 82.30 |

Table 1: **Amodal Completion on KINS**. Invisible mIOU means we evaluate mIOU only on invisible areas. GT Crop denotes that input is cropped by GT amodal bounding box.

| Method | Full mIOU | Invisible mIOU |
|---|---|---|
| Amodal-VAE | **94.68** | **62.85** |
| w/o. Full-Mask training | 94.28 | 58.92 |
| w/o. Simulated training | 83.30 | 35.82 |
| w/o. Likelihood training | 94.04 | 57.04 |
| - Latent Space L2 loss | 94.02 | 56.90 |
| w/o. Class Conditioning | 93.56 | 53.03 |

Table 2: **Ablation study** of Amodal-VAE on KINS.

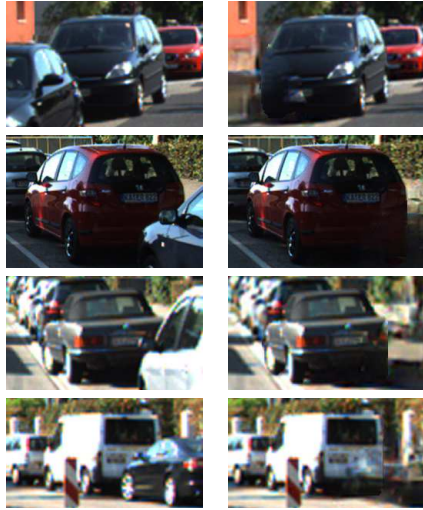

Table 3: Examples showing our shape and appearance completion on KINS. **Left:** before completion; **Right**: after completion

compute the cosine similarity between an input partial mask and the synthetic partial masks and then use as output the full mask corresponding to the synthetic partial one with the highest similarity to the input. Results show that Amodal-VAE outperforms this baseline (*Nearest Neighbor Mask* in Table 1).

We further ablate the use of the RGB information as additional input to the VAE. After the full-mask-only training stage, we use a ResNet-50 pretrained on ImageNet, which takes cropped RGB images as input, concatenate the ResNet's features and the mask encoder output, add two further convolutional layers to merge the two, and predict the latent code posterior distribution. The ResNet is finetuned together with all other trainable parameters and we optimize the setup's hyperparameters and report the best result. As shown in Table 1, line *RGB-Amodal-VAE*, the additional RGB-based image features do not boost performance. Hence, for our main Amodal-VAE model we discard the RGB input for simplicity. It is possible that a more carefully designed model architecture will be able to extract more useful information from the RGB input as the slight decrease in performance might seem counterintuitive, but we leave this for future research.

In the experiments above, we always tightly crop the instance mask. However, in an interactive scene editing tool, users can be asked to provide the amodal box. Thus, we evaluate our method also by utilizing GT amodal bounding boxes, which precisely indicate the extent of the occluded area. In these experiments (*Amodal-VAE + GT Box*), we achieve 97.64% and 82.30% mIOU, respectively. This suggests that there is much room for improvement by better cropping the input masks automatically.

**Posterior sampling:** To further motivate the use of a probabilistic model, we show quantitative results from multiple posterior predictions. For each partial mask instance, we sample 20 latent codes from the approximate posterior distribution and decode to the corresponding completed masks. We calculate mIOU using masks with the best visible area IOU or best amodal GT IOU. The results in Table 4 show that by sampling we find masks that match the amodal GT significantly better than using the approximate posterior mode. Hence, the approximate posterior incorporates diverse plausible masks, correctly capturing the ambiguity. Using samples from the full posterior distribution may benefit downstream applications. Additional results are provided in the supplementary material: We analyze approximate posterior widths as a function of occlusion ratio and we also show prior samples.

**Ablations:** We first ablate the three training stages described in Sec. 4.1. The results in Table 2 show that the performance drops by 3.93% if we omit the first Full-Mask-only training stage. Furthermore, the model performs significantly worse without the second occlusion-simulation training stage, because this is where the model learns to actually map partial to full masks. Likelihood-based (i.e. using the ELBO) partial-mask-only finetuning as the third stage plays an important role, since it brings real occluded instances into the training loop. Also, conditioning on class information is crucial, as it helps the VAE to better infer the masks, especially when there is a large occlusion.

Next, we conduct cross dataset evaluations. We train Amodal-VAE on the Cityscapes training set and evaluate on the KINS test set. Due to the mismatch in class categories across datasets, we merge the bus and car classes into one class, and motorcycle and bicycle classes into another. Results in Table 5 show cross domain stability of our model. We consistently outperform the De-occlusion baseline.

| Method | Full mIOU | Invis. mIOU |
|---|---|---|
| Amodal-VAE | 94.68 | 62.85 |
| Search by vis. Mask | 94.71 | 62.98 |
| Search by amodal GT | 95.82 | 69.96 |

Table 4: **Mask completion results for predictions using multiple posterior samples.** We sample masks from approx. posteriors and search for best samples via visible area or full amodal GT area IOU. "Amodal-VAE" uses only the posterior mode.

| Method | GT Crop | Full mIOU | Invis. mIOU |
|---|---|---|---|
| Amodal-VAE | ✗ | 93.72 | 56.18 |
| De-occlusion | ✗ | 93.19 | 48.23 |

Table 5: **Cross Domain Amodal Completion**. Models are trained on Cityscapes and tested on KINS.

| Amodal-VAE GT-box | Ground Truth | No Preference |
|---|---|---|
| **46.68** | 39.50 | 13.8 |

Table 6: **User Study**. We evaluate our model against human-annotated amodal masks in KINS via an Amazon Mechanical Turk user study. Interestingly, subjects prefer our object completions to the human-labeled ones.

**Qualitative results:** We show qualitative results in Figure 3. We also compare to human-annotated masks in Figure 4. Our generated masks contain more details and look more natural than GT masks. We further show shape variations by sampling from the approximate posterior distribution in Figure 6 and Figure 7. Different plausible completions are drawn from a single partial mask.

**User study:** We also evaluate our model against human-annotated amodal masks in KINS via an Amazon Mechanical Turk user study. We assume that the user draws the amodal box which is provided to Amodal-VAE. We randomly sampled 3260 instances from the KINS test set and asked Turkers to indicate preference between Amodal-VAE's amodal masks and GT annotated amodal masks. Interestingly, as shown in Table 6, users prefer Amodal-VAE's masks **46.68%** of the time versus 39.50% for ground truth. This demonstrates that Amodal-VAE outperforms the drawing skills of the human annotators of the KINS dataset [27].

In the supplementary material, we provide additional results on amodal segmentation, where we first predict modal segmentation masks using a standard segmentation model, and then use Amodal-VAE to complete partial segmentation masks.

### 5.2 Object Manipulation Application

Here, we apply the Amodal-VAE to interactive scene editing and report the results.

**Background and Instance Inpainting:** Since Amodal-VAE can be used to predict complete instance masks for all objects in a scene, we can use these inferred masks to move or delete objects. Such operations will uncover previously occluded parts of the objects and the background. We complete the missing content using an inpainting neural network, which takes RGB images with missing content as input and generates a realistic completed output. Similar to [39], we are using the convolutional inpainting network from [25], which employs partial convolutions and nearest neighbor up-sampling in the decoding stage. Inpainting details are available in the supplementary material.

We benchmark the performance of instance inpainting. Since we do not have any ground truth appearance for the invisible areas, we exploit Fréchet Inception Distance [12] (**FID Score**) to evaluate the inpainting results. FID is a measure of similarity between two datasets of images. It was shown to correlate well with human judgment of visual quality and is most often used to evaluate the quality of samples from Generative Adversarial Networks. FID is calculated by computing the Fréchet distance between two Gaussians fitted to feature representations of the Inception network [33]. In our case, we use non-occluded instances in the KINS test set as a reference dataset. For each instance, we use Amodal-VAE and the inpainting network to complete the mask and appearance. We compute FID distances between the reference dataset and inpainting results based on predicted amodal masks. Intuitively, the better and more natural the amodal mask is, the lower the FID score should be. Our Amodal-VAE achieves **41.44** versus 50.36 for the baseline De-occlusion approach. Note that the inpainting networks we use for both methods are identical. We thus conclude that the amodal masks predicted by Amodal-VAE lead to more natural completions.

**Instance Manipulation:** Furthermore, we show how we can change the pose of objects, even of those which are partially occluded. Since we are working with complex street scenes, we focus on cars for this demonstration. We exploit GauGAN [26], which can separately take into account local appearance and mask shape. We first infer an object's complete shape and appearance as described above. Then, we use GauGAN's encoder to infer its latent representation, which captures only local appearance information. When regenerating the image, we randomly sample complete shapes from the test set and feed them to the SPADE layers. Due to the separation of local appearance and global semantic information in GauGAN, the newly generated scene reflect any pose changes.

**Qualitative results:** We first show the qualitative inpainting results in Table 3. Conditioned on the complete mask, the inpainting network recovers the invisible appearance successfully. We also

|   |   |   |   |
|:---:|:---:|:---:|:---:|
| Image | Swap Order | Move & Scale | Change Pose |

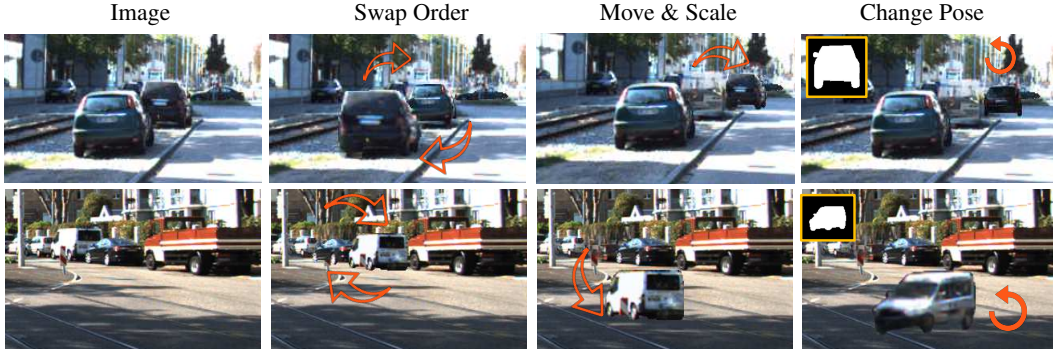

Figure 5: We demonstrate several object manipulation options in our interactive scene editing tool. We first perform amodal mask completion and inpainting of the original object, then manipulate instance size and position. We also rely on GauGAN to generate the pose-changed instances.

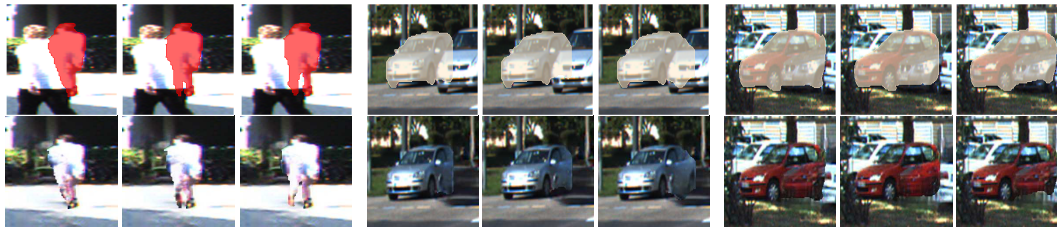

Figure 6: We sample masks from the approximate posterior distributions, which leads to different inpaintings in invisible areas. Results are shown on KINS dataset.

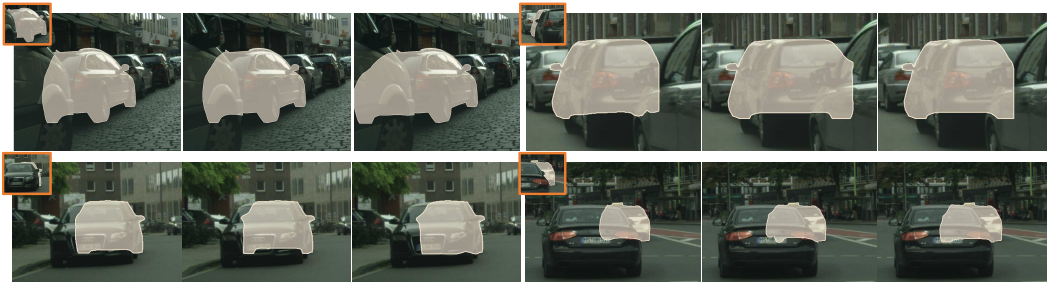

Figure 7: We sample masks from the approximate posterior distribution. Results shown on Cityscapes dataset.

deleted the foreground mask by another background inpainting module. In Figure 5, we showcase different functionalities in our interactive scene editing tool. Based on the amodal mask, our tool supports swapping order, deleting, moving, and scaling objects. We also showcase how we can change the pose of objects by utilizing GauGAN as described.

# 6   Conclusions

In this work, we propose Amodal-VAE, a simple probabilistic method for amodal instance completion, which does not require amodal labels for training. In particular, our method is based on a variational autoencoder that learns to reconstruct full object masks from partially occluded ones by using a carefully designed training strategy. This exploits both full and partial instances available in existing segmentation datasets. We quantitatively and qualitatively showcase the performance of our method on the downstream task of scene editing of complex street scenes. Our experiments show significant improvement over the recently proposed state-of-the-art method. We provide our method as an interactive image editing tool where users can remove, move, or swap different objects in the image.

Note that training Amodal-VAE requires a high quality dataset with complete masks and each category must contain a sufficient number of objects. Therefore, in this work we focus on driving scenes which contain mainly rigid objects and for which sufficient data is available. Applying our model on more complex scenes and in a setting with limited data is left for future work.

# 7 Broader Impact

Our proposed model can be used in a wide range of applications that require reasoning on occluded objects. These include planning tasks in robotics, object tracking, and editing a photo or video. We focus on two significant impacts of using our model. The first is in the context of autonomous driving. An autonomous driving car must infer the geometry and identity of surrounding objects for its decision-making process. Partially visible objects could lead to wrong estimates for motion planning, and thus reasoning about the full extent of objects can lead to much safer control. Our approach infers the complete shapes of the occluded objects for this purpose. The other major impact is on augmented reality. One could use our technology to snap a photograph of their environment, and "delete" existing objects from the photograph, replacing them with alternatives. The crux of our approach is in deleting content from an image, which could be subject to misuse. We encourage work on detecting fakes as the standard technology to deal with image manipulation approaches.

## Acknowledgments and Disclosure of Funding

This work was fully funded by NVIDIA and no third-party funding was used.

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
