[Supplementary Material]

# Supplementary Material:
# Variational Amodal Object Completion

**Huan Ling**[1,2,3]    **David Acuna**[1,2,3]    **Karsten Kreis** [1]

**Seung Wook Kim**[1,2,3]    **Sanja Fidler**[1,2,3]

NVIDIA[1]    University of Toronto[2]    Vector Institute[3]

{huling ,dacunamarrer, kkreis, seungwookk, sfidler}@nvidia.com

## 1   Amodal-VAE Analysis

In this section, we further analyze Amodal-VAE.

We show generative samples from the model in Figure 3. We randomly sample from the factorial Normal prior distribution $p(z) \sim \mathcal{N}(0, \mathcal{I})$ and transform the samples into masks using the decoder $p_{w_1}(y|z)$. The masks obtained by sampling from the prior resemble diverse car-like and pedestrian-like shapes. This validates that Amodal-VAE correctly learnt a generative model of realistic instance masks during training stage (1).

Next, we explore how Amodal-VAE captures the ambiguity inherent in the mask completion task. Intuitively, we expect the approximate posterior distribution to be wider when the partial input mask corresponds to a very occluded object and to be narrower when most of the object is visible. Since our approximate posterior is also modeled using a factorial Normal distribution, this can be tested easily. We calculate the standard deviations predicted for all samples from the KINS test set, which we partition into three categories based on the amount of occlusion. The results are shown in Figure 4 and validate our intuition. The more occluded an object is, the wider is the corresponding predicted approximate posterior distribution in latent space. In other words, Amodal-VAE correctly models shape uncertainty caused by occlusion. It is more certain when completing masks of mostly visible objects, but it also captures the increased uncertainty when completing masks of very occluded objects. We can make use of this and calculate different plausible completions by sampling from the approximate posterior and decode to the corresponding masks (as shown in Figures 6 and 7 in the main text and Figure 9 below).

Furthermore, we report mIOU versus occlusion percentage in Figure 1. For the *Amodal-VAE best in samples* curve, we sample 20 latent codes from the approximate posterior distribution and decode to the corresponding completed masks. We then calculate mIOU using masks with the best amodal GT IOU. This is a similar experiment like the posterior sampling experiments shown in the main paper. The results demonstrate that by sampling we can find better masks than using the approximate posterior mode. Importantly, the gap between the curves narrows as the occluded area becomes smaller. This is in line with the previous analyses, since less occlusion implies less ambiguity in the mask completion task. Hence, the approximate posterior distributions are narrower and we profit less from sampling.

## 2   Amodal Segmentation

We provide additional experiments on amodal segmentation using the KINS dataset in Table 1. We crop the images by GT bounding box and train a ResNet-PSP segmentation model [2, 7]. Then, we predict the instance masks on the test set, yielding 80.83% amodal mIOU. We use the predicted instance masks as input to perform amodal completion using both our Amodal-VAE and the baseline, which we outperform. We also find that our model is robust to instance mask corruptions. Even

though full mIOU drops by 8.65% (94.68% vs. 86.03%, also see Table 1 in main text), mIOU on invisible area only drops by just 2.14% (62.85% vs. 60.71%).

| Method | Full mIOU | Inv. mIOU |
|---|---|---|
| GT Instance Mask | 87.03 | 0 |
| Pred. Mask | 80.83 | 0 |
| Amodal-VAE | **86.03** | **60.71** |
| Deocclusion | 84.94 | 52.93 |

Table 1: **Amodal Segmentation on KINS**. Models take predicted partial segmentation masks as input.

Figure 1: **Amodal Segmentation mIOU with different occlusion ratios on KINS.** X-axis represents visible area ratio.

## 3 Implementation Details

In this section, we provide implementation details. All models are trained on Nvidia Tesla V100 GPUs.

### 3.1 Training Details:

We train all three steps described in Sec 4.1 of the main text until the model converges on the validation set. Following [6] and for a fair comparison, we use the validation set to search for an optimal threshold $t$ to use when calculating binary masks from the probabilistic output of the decoder $p_{\boldsymbol{w}_1}(\boldsymbol{y}|\boldsymbol{z})$. This resulted in a threshold $t = 0.1$, which we use to complete masks at test time.

We train the model using the Adam optimizer [3] with learning rate $10^{-3}$ in the first training stage and $10^{-4}$ in the second and third stages. To prevent posterior collapse during training in the first stage, we slowly anneal the KL weight $\lambda$, stopping at 0.5 [1]. This improved the performance on the mask completion task. Specifically, we implement $\lambda$ as a function of training step $t$:

$$\lambda(t) = \lambda_{final} - (\lambda_{final} - \lambda_{start}) \times \alpha^{max(0, t - t_{begin})} \tag{1}$$

We use $\lambda_{final} = 0.5$, $\lambda_{start} = 0.01$, $\alpha = 0.9995$ and $t_{begin} = 2000$.

### 3.2 Amodal-VAE Implementation Details:

**Model Design:** The input and output masks of Amodal-VAE are tightly cropped and resized to 28×28 resolution. As shown in Figure 2, both encoder and decoder are convolutional neural networks

followed by two residual blocks and fully connected layers. We use ReLU activation functions and batch normalization, which we apply before the activation functions. As discussed in Section 4.2 in the main text, the input and output correspond to different scales with respect to the underlying image. This is corrected using a spatial transformer, which is also parametrized as a convolutional neural network with max pooling. Our latent space is 64-dimensional.

### 3.3 Inpainting implementation details

**Background Inpainting:** We inherit implementation details and code from [4]. Similar to [6], we randomly crop 256×256 patches from the image as input. We simulate background holes by randomly placing instance masks on patches. We randomly dilate instance masks from 5 to 25 pixels.

**Instance Inpainting:** We slightly modify [4] to accept cropped instances as input. We resize instance images to 256×256 resolution. At training time, based on the full mask data and RGB images, we simulate various occlusions by randomly placing one instance onto another, hence generating a synthetic dataset of paired RGB images with holes and complete RGB images. We fill the holes with the corresponding class label as the final input to the inpainting model. At test time, given a full mask generated by Amodal-VAE, we create holes according to the invisible area of the completed full mask.

### 3.4 GauGan Implementation Details:

We inherit implementation details and code from [5]. We slightly modify the code and provide instances as input, such that the inputs to the SPADE layers are single-channel full masks. We train GauGan based on full masks and the corresponding RGB images for 200 epochs.

## 4 Additional Qualitative Results

In this section, we provide additional qualitative results. We visualize and compare Amodal-VAE predictions and human-annotated amodal masks in Figure 5. Figure 6 demonstrates background and instance inpainting when removing foreground objects and completing shapes of previously occluded objects. We also show failure cases of Amodal-VAE in Figures 7 and 8. Notice that most failure cases are due to incorrect bounding box predictions and can be solved by asking humans to draw amodal bounding boxes. We show more posterior sampling results in Figure 9. We also show how we can use GauGAN [5] to change instances' poses in Figure 10.

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

Figure 2: Architecture of Amodal-VAE.

Figure 3: Samples from the prior, decoded to masks before binarization. The greyscale pixel values indicate Bernoulli probabilities $\in [0, 1]$.

Figure 4: We demonstrate that the width of the approximate posterior distribution depends on the amount of occlusion in the partial mask input to Amodal-VAE. We are looking at the 15 latent variables with the smallest predicted approximate posterior standard deviations. For each, we are calculating the average predicted standard deviations for partial masks from the KINS test set when partitioned into three categories: Cases with without any occlusion (blue). Cases with 1-50% of the object being occluded (orange). Cases with more than 50% of the object being occluded (green). We see that a higher level of occlusion implies a wider approximate posterior, demonstrating that Amodal-VAE correctly captures the ambiguity in the mask completion task. Note that the last 3 latent variables shown have standard deviations $\approx 1$, independent of occlusion level. Hence, they match the prior and are effectively turned off. All other latent variables that are not shown are similarly turned off.

**GT**                                    **Pred**

Figure 5: Predicted mask completions of Amodal-VAE (Pred) vs human-annotated ground truth (GT) amodal masks. Results are shown on the KINS test set.

| Instance Mask | Amodal Prediction | Before Inpainting | After Inpainting |

Figure 6: Examples showing our shape and appearance completion on KINS. We remove a foreground instance, complete the previously occluded object, and inpaint the previously invisible part of the object as well as the background.

Figure 7: Examples of Amodal-VAE failure cases when completing masks. The failures can be solved by providing the ground truth (GT) amodal box.

Figure 8: Examples of Amodal-VAE failure cases when completing masks. The failures can be solved by providing the ground truth (GT) amodal box.

Figure 9: We complete partial masks by decoding different approximate posterior distribution samples. Results shown on Cityscapes dataset.

Figure 10: We demonstrate that we can generate instances with modified poses, relying on GauGAN.