[Reviews · NeurIPS 2020]

Review 1

Summary and Contributions: The paper proposes a method based on VAE for amodal object completion without ground truth amodal segmentation annotation (achieve via occlusion simulation). The paper also demonstrates application including impainting and instance manipulation.

Strengths: The paper proposes a sound method based on established VAE model for occluded shape encoding and full shape generation. Although the VAE modeling itself is not new (similar to [30]), the paper is able to demonstrate its applicability to the domain of shape segmentation via both qualitative and quantitative evaluation, as well as downstream applications.

Weaknesses: [1] Limited novelty. As mentioned above, the key components of the method has mostly been covered in [30] with the major difference being the **domains** both papers address (3D shape completion for [30] and instance segmentation completion for this paper). That being said, the theoretical contribution of this paper is limited. In addition, the occulusion simulation as a way to generate paired occluded shape-full shape is already well utilized. [2] Lack of comparison to the baseline. There is only one baseline that is compared against in the paper, which is limited. Also a lack of qualitative comparison between the method and the baseline weakens the conclusion that the method is superior not only numerically but also visually. Also the paper only evaluates on the car category of KITTI. Is it possible to evaluate on multiple categories of different amodal shape datasets? As we know the car category is a relative easy one, as cars shapes do not share a lot intra-class variation compared to some other categories. For example, as the method only uses occluded shape as input, it is possible to create synthetic datasets from ShapeNet by rendering into 2D silhouettes, similar to [30]? [3] Further clarification. How are the spatial transformation network learned? What is the supervision? [4] Although demonstrating the downstream application of the method in Section 5.1 is much appreciated as it provides insights into how useful the model is, this section is not very relevant to the central task -- amodal mask completion, because such applications can be done on all methods in this task, and a lack of comparison with baseline method had failed to demonstrate how the proposed method overtakes prior arts in those applications.

Correctness: Yes the claims of contribution as well as the methodology are well evaluated in the paper.

Clarity: Yes the paper is generally welll-written, except for some points to clarify (see Weakness 2 and 3).

Relation to Prior Work: Yes the prior arts and how this paper differentiates from them is detailed in the Related Work section.

Reproducibility: Yes

Additional Feedback: It would be better if the author can provide any insight into why there is only one baseline in comparison. Also the evaluation is limited on which it is hard to evaluate the full performance of the method. === Post rebuttal Firstly I would like to thanks the authors for addressing my concerns! However my major concern still holds as the lack of novelty in the paper. As far as I am concerned all modules in the method has appeared in related tasks (e.g. 3D shape completion is not inherently different from 2D mask completion; where the difference int the models may just be 2D encoder-decoder VS 3D ones, which is a trivial change), and also the key idea of generating synthetic training pairs for this tasks appeared before as occlusion simulation.


Review 2

Summary and Contributions: This paper studies the task of amodal object mask completion. Given the mask of an occluded object, the paper learns a network to complete the mask. To account for the ambiguity in such mask completion, the paper pursues a variational approach. The approach is trained in a stage-wise manner on a combination of occluded and un-occluded object masks. The paper conducts evaluation on the KINS dataset, and shows the effectiveness of the various proposed components, as well as past methods on this task.

Strengths: The paper is well-written. Experimental evaluation uses latest datasets for the problem being studied.

Weaknesses: 1. Limited novelty, The paper applies and adapts standard ideas in the field for the application of amodal mask completion. As such, the paper has limited novelty. Furthermore, the application being considered is only mildly interesting (more on this in the next point). 2. While amodal instance completion (predicting the occluded pixels) is an interesting problem. The problem of predicting the amodal instance segmentation from partially occluded object is also interesting. The problem of predicting the full mask from a partial mask, in my view, is less interesting. All applications that the paper considers, have the appearance of the occluded object available, and it is a natural question as to why that information is being thrown away. Thus, I find the application inadequately motivated. In fact, if the appearance in the occluded mask is taken into account, it is also possible that the problem in fact simplifies -- there is likely less ambiguity when appearance is taken into account, than when it is not. 3. No quantitative evaluation for prediction of different modes. The paper motivated the use of a variational approach, so as to be able to make multiple predictions in case of ambiguities. The paper only presents a few qualitative examples.

Correctness: Looks correct

Clarity: Paper is well written.

Relation to Prior Work: Paper presents a thorough related work survey.

Reproducibility: Yes

Additional Feedback: == Post-Rebuttal == Thanks for your response, and for providing additional experiments. I still feel that the total novelty (technique + application) is low. Rebuttal states that proposed approach is an application of existing methodology (thus, not novel). The paper would still have been worth accepting if the application being tackled was interesting, and experiments were thorough and brought out new insights. In my view, this is not the case. I don't know what new knowledge I should take back from the paper. Rebuttal claims a novel training strategy to be a contribution. I don't find the novel training strategy principled. It comprises of multiple stages of training, which as far as I understand, requires a dataset of amodal masks.


Review 3

Summary and Contributions: The paper proposes a variational autoencoder for estimating the complete mask of a partially occluded object from its visible mask (segmentation) in an image. Through a three stage training process reminiscent of curriculum learning the method is able to learn mask completion without relying on a paired dataset with human annotated completions. The method is evaluated on objects appearing in typical street scenes (cars, bikes and pedestrians).

Strengths: + The paper is very well written and easy to follow. The related work and background is well described. + The three stage training process has good motivation (and perhaps can be thought of as a type of curriculum learning). + The results appear to be of practical use (in image editing) and user studies demonstrate the advantage of the approach.

Weaknesses: - The method requires additional information to what is typically provided in a semantic segmentation dataset in that it needs to know whether masks are complete or incomplete for training. - While the ablation study shows the importance of the unpaired data for finetuning but also highlights the fact that the invisible component of the objects are typically small (i.e., far fewer pixels than the visible part). It would be interesting to see a plot of accuracy versus percentage occlusion. - The multiple plausible completions is never really tested given that the experiments are only run on rigid objects (I'm considering pedestrians here as rigid given their size and viewpoint in the chosen datasets). It would have been interesting to see results on articulated objects (such as animals) or objects where there is a greater variation in possible object shape.

Correctness: The method is empirical. It appears to follow standard practice in the field. A user study (human evaluation) is performed but full experimental details are missing. For example, how many users? How many repeated trials? Were the users trained on the task? Etc.

Clarity: The paper is very clear and easy to follow.

Relation to Prior Work: Related work and background material is presented well.

Reproducibility: Yes

Additional Feedback: Minor comments: - The title should be changed to "Variational Amodal Object Mask Completion" to highlight the fact that the contribution is in completing the binary object mask rather than inpainting the occluded part of the object. - The statement on Line 25 is slightly misleading since (bounding box) object detection and articulated parts models (such as human skeleton) do attempt to infer the location of hidden/occluded parts. The statement is true of semantic segmentation research. - Line 102: "rigorous" -> "strict" (perhaps?) - Section 3. Is \mathbb{Y} a strict subset of \hat{\mathbb{Y}}? The definition of these spaces could be made more clear. - Line 137: \mathbb{Y} and \hat{\mathbb{Y}} appear to be switched here. POST REBUTTAL: I am still concerned about the variability learned by the model. It seems to be able to handle a very small number of rigid categories. More work needs to be done to demonstrate the model on a wider range of object categories including articulated objects. The paper is marginally above the acceptance threshold.


Review 4

Summary and Contributions: This paper tackles two problems in amodal completion: the difficulty and ambiguity of occlusion annotation. It proposes a VAE framework that trains on unpaired full masks and occluded masks in three stages, so that encodings of full and occluded masks are learned and aligned, and decoding of full masks from the aligned latent space is also learned. The method is used for mask completion, inpainting, and scene manipulation.

Strengths: Strengths: - The method is simple yet effective, relieving supervision demand for mask completion (not sure for the first time or not). - The three-stage VAE training is interesting and intuitive, and hopefully is able to inspire methods in other tasks. - Mask completion results look good, and hand in hand with other tools, the method can also be applied to inpainting and multiple scene manipulation tasks.

Weaknesses: Weaknesses: - Some masks do not look reasonable when visual cues are considered, for example, the top left mask of Figure 7, which nevertheless might make sense without the image shown. Despite the paper saying "due to the nature of our Amodal-VAE, we discard RGB pixels...", I wonder if the VAE is also able to condition on the instance appearance somehow and if it helps. * Rebuttal experiment shows adding RGB cannot help. It didn't surprise me much, as it makes training input more "noisy" and training more easily overfitting (to some RGB features). Humans can leverage RGB in sort of a reasoning way, i.e. when the mask can have two explanations, use RGB to match the two hypothesis via some mental simulation, and decide. This can be too hard for neural networks trained for one task. So I think this part helps justify the methodology. - Empirically, full masks and occluded masks of cars follow strong patterns in datasets like KITTI. In other domains, the lack of visual cues might hurt more. More results from different instance classes, or more comparisons with methods that take RGB inputs [1,2,3] or even 3D cues [4], would be valuable to address my concerns. * Rebuttal: I can also understand the object category is limited by dataset availability. Glad if author "show quantitive results on more classes and non-rigid objects for the camera-ready version" as claimed. - For the scene manipulation part, the pipelined approach does not integrate visual or 3D cues in the Amodal-VAE part, and GauGAN seems to lift a heavier job. Also, no comparison or quantitive results are shown. To this end, [5] provides a scene manipulation benchmark and a baseline to compare with (via user study or metrics like FID). [1] Amodal instance segmentation. ECCV 2016. [2] Amodal instance segmentation with kins dataset. CVPR 2019. [3] Visualizing the invisible: occluded vehicle segmentation and recovery. CVPR 2019. [4] Learning 3d shape completion under weak supervision. IJCV 2018. [5] 3D-aware scene manipulation via inverse graphics. NeurIPS 2018.

Correctness: Method and experiments do not have major flaws.

Clarity: The paper is clearly written, with good math formulations and intuitive motivations.

Relation to Prior Work: Yes.

Reproducibility: Yes

Additional Feedback: - Isn't Table 4 a figure? - Ambiguity plays a major role when the occluded mask is small (e.g. Figure 7 bottom), but not for big occluded masks (e.g. Figure 7 top). I wonder if some uncertainty control can be used based on mask size.

[Author Response · NeurIPS 2020]

| Method | RGB | Full mIOU | Invisible mIOU |
|---|---|---|---|
| Amodal-VAE | ✗ | **94.68** | **62.85** |
| Deocclusion | ✓ | 94.04 | 57.19 |
| ResNet-Amodal-VAE | ✓ | 94.53 | 61.97 |

Table 1: **Amodal Completion on KINS**. Checkmark on RGB column means the RGB image is used as input to the model.

| Method | Full mIOU | Inv. mIOU |
|---|---|---|
| GT Instance Mask | 87.03 | 0 |
| Pred. Mask | 80.83 | 0 |
| Amodal-VAE | **86.03** | **60.71** |
| Deocclusion | 84.94 | 52.93 |

Table 2: **Amodal Segmentation on KINS** Models take predicted segmentation mask as input.

| Method | Full mIOU | Invisible mIOU |
|---|---|---|
| Amodal-VAE | 94.68 | 62.85 |
| Search by visible area | 94.71 | 62.98 |
| Search by amodal GT | 95.82 | 69.96 |

Table 3: **Results for multiple predictions using samples.** We sample masks from approx. posteriors and search for best samples via visible area or full amodal GT area IOU. "Amodal-VAE" uses only the posterior mode.

Figure 4: **Amodal Segmentation mIOU with different occlusion ratio on KINS.** X-axis represents visible area ratio. Better in zoomed-in view.

We would like to thank the reviewers for their constructive feedback.

**R1&R2: Novelty.**    Although methodologically related approaches have appeared in the literature before, we focus
on amodal object mask completion where probabilistic methods have not been explored. For this task, we are the first to
propose a generative framework, which naturally captures the ambiguity inherent in the mask completion. Since naively
applying a VAE is not possible, we propose a novel training strategy. Compared to previous work that uses VAEs for
3D shape completion [30], we further perform occlusion simulation and bounding box prediction, add latent code
regularization during training. These contributions allow us to (1) achieve state-of-the-art results on the amodal mask
completion task without amodal ground truth supervision. (2) Our mask completions outperform the drawing skills of
human annotators. (3) We demonstrate the usefulness of our model on the downstream task of scene editing. (4) We
qualitatively and quantitatively (Tab. 3, Fig. 4 above) show the value of explicitly modeling the uncertainty in the task.

**R2&R4: Will RGB input make the task trivial?**    We slightly modify the model architecture and provide additional
results. After the full-mask-only training stage, we use a ResNet-50 pretrained on image-net and taking RGB images
as input. We concatenate the ResNet's features and the mask encoder, add two convolution layers, and predict latent
code posterior distributions. As shown in Table 1, the additional RGB-based image features do not boost performance.
Hence, the mask together with its object class contains enough information and we discard RGB input for simplicity.

**R2: Motivation of amodal completion & Amodal instance segmentation is more interesting.**    We agree that
amodal instance segmentation is also an interesting problem. However, amodal instance segmentation can be decom-
posed into instance segmentation and amodal completion. We provide additional experiments on amodal segmentation
(results in Tab. 2) and will update the paper. We crop images by GT bounding box and train a ResNet-PSP segmentation
model. We predict the instance masks on the test set, yielding 80.83% amodal mIOU. We use the predicted instance
masks as input to perform amodal completion using both our model and the baseline (Deocclusion [37]), which we
outperform. We also find that our model is robust to instance mask corruptions. Even though full mIOU drops by 8.65%
(94.68% vs. 86.03%), mIOU on invisible area only drops by just 2.14% (62.85% vs. 60.71%).

**R2: Lack of quantitative results for multiple posterior predictions.**    We agree and quantitatively evaluate this
aspect (paper will be updated). For each instance, we sample 20 latent codes from the approx. posterior distribution.
We calculate mIOU using masks with the best visible area IOU or best amodal GT IOU. Results show that by sampling
we find masks that significantly better match amodal GT than using the approx. posterior mode (Tab. 3 above). Hence,
the approx. posteriors incorporate diverse plausible masks, correctly capturing the ambiguity. This suggests the use of
multiple posterior samples in downstream applications. Sampling is more beneficial for cases with significant occlusion
(Fig. 4 above) and we also show (Fig. 3 of supp. material) that more occlusion correctly results in wider posteriors.

**R1&R3&R4: Results are only shown on KINS dataset.**    We primarily focus on driving scenes and exploit two
datasets with multiple classes, KINS and Cityscapes. KINS contains 7 categories (pedestrian, cyclist, person sitting, car,
van, tram, truck) and Cityscapes contains 8 categories (bicycle, bus, person, train, truck, motorcycle, car, rider). We
show quantitative results only on KINS, because it is the only available large-scale amodal dataset with accurate human
annotations. We plan to show quantitive results on more classes and non-rigid objects for the camera-ready version.

**R1 & R4: Comparisons for downstream tasks.**    We appreciate the suggestions. However, our main contribution is
on the amodal completion task. Comparisons to other downstream task methods are beyond the scope of our paper.
Nevertheless, we compared FID scores with Deocclusion [37] on the image editing task.

**R1: Need to compare with more baselines & Only one category.**    We apologize for the confusion and will clarify
this aspect in the camera-ready version. However, all experiments are conducted on datasets with multiple classes. We
use Deocclusion [37] as baseline, since it is the current state-of-the-art model on amodal completion with no amodal
ground truth supervision. Also, our user study demonstrates that we outperform the annotation skills of humans, which
further strengthens our conclusions. We will add additional details about the user study in the final version of the paper.

**R1: Transformation network.**    The spatial transformation network is end-to-end differentiable and trained as part of
the pipeline in stages (2) and (3), using the mask reconstruction loss as supervision (we will clarify in the final version).

**R3: Accuracy versus occlusion percentage .**    We agree. We show results in Fig. 4

**Details, etc:**    We appreciate the suggestions and will incorporate them in the camera-ready version.

[Meta-Review · NeurIPS 2020]

This submission tackles the problem of amodal category-specific instance mask completion. To do this, they propose an interesting 3-stage training process for a variational autoencoder that maps partial masks to full masks, followed by resizing to match the object sizes. Reviewers were divided on whether the curriculum training process represents an important contribution; I think this is well-designed, but it could be more clearly motivated in the text. This is demonstrated both for the mask completion problem, and through combination with instance inpainters, for the instance completion problem in the RGB pixel space. During rebuttal experiments, authors also showed results (Tab 3, Fig 4) indicating that the method is able to produce diverse predictions in the occluded regions. While not all reviewers were convinced on this point, I found this result very helpful in evaluating the usefulness of the probabilistic predictions. The fact that the deocclusion results are better than [37] with a simpler approach is also an important strength. This paper could still improve significantly in the camera-ready. An important drawback pointed out in original reviews whose addressal in the rebuttal is a bit contentious among the reviewers is: why should mask completion be constrained to not use RGB information. The author response, somewhat unintuitively, shows a small drop in performance from using RGB data, but this is one specific instantiation of how RGB information might be used, and it does not rule out that RGB information is in fact valuable using a different approach. Another drawback that the authors have promised to address (but is pending now) is the fact that the number of categories evaluated on are somewhat limited in the manuscript. Finally, I would also suggest a few improvements: (i) a nearest neighbor baseline that completes masks by matching to the closest mask in the synthetic dataset, (ii) an additional ablation of the pixel reconstruction loss in stage 2 to answer: does the latent space loss alone suffice, (iii) explicitly show results demonstrating that separating training stages 1 and 2 does actually improve performance. (iv) discuss hyperparameters such as training lambdas for various loss terms, and finetuning schedule etc.